# Social Services Management in the Context of Ethnic Roma Issues in the Czech Republic with a Focus on Education for Roma Children

**Marek Merhaut [1], Marie Fulkova [2] and Lothar Filip Rudorfer [3,*]**

1   Department of Science and Research, Faculty of Education, Charles University, M. Rettigové 4, 110 00 Prague, Czech Republic

2   Department of Art Education, Faculty of Education, Charles University, M. Rettigové 4, 110 00 Prague, Czech Republic

3   Department of Psychology, Faculty of Education, Charles University, M. Rettigové 4, 110 00 Prague, Czech Republic

*   Correspondence: lotharfilip.rudorfer@pedf.cuni.cz

**Abstract:** According to qualified estimates by sociologists, the Czech national census and governmental bodies in the EU, as many as 350,000 ethnic Roma people live in the Czech Republic. Yet, only a fraction of the Roma population has received higher education and are achieving careers within and not alongside the majority society. Some believe that a more inclusive and integrative approach in education will facilitate long-term community work and better removal of prejudices. This represents quite a force that could appreciably affect the political landscape. Unfortunately, we run into the disunity of the Roma people among themselves, which brings more challenges to the forefront rather than solutions to the current status quo. This should change in the future. The current government concept for this purpose supports the development of Roma culture and education. This overview study should bring more arguments to the table regarding the need for Czech Romani/Roma intervention in education and social affairs. This overview article gives an insight into current Czech Roma/Romani education and social service-related issues.

**Keywords:** social issue; Roma ethnic; Roma children; education; management of social issue; racism; political landscape; migration; segregation; inclusion; social exclusion

## 1. Introduction

Addressing the issue of social exclusion of Roma people using social work tools in the Czech Republic and the EU is very difficult, as social work only confronts some symptomatic manifestations, but does not eliminate the deep causes of social exclusion, as previously investigated by Scullion and Brown (2017). The Czech Ministry of Labor rejects the claim that members of the Czech society who object to the current arrangements and so-called solutions to the Roma issue are racists. Sivic et al. (2013) argue that, on the contrary, there is a need to look for reasons for the current failure of past social programs, and criticism of government programs cannot be confused with racism under the pretext of false humanism. On the other hand, we must acknowledge that complex problems do not have simple solutions, and it is impossible in a democratic society in the long term to allow promises of easy solutions to the Roma issue to become a lift to political posts—simply because, from a systemic perspective, those promises cannot be fulfilled, as stated by Urh (2014). From the suggested point of view, there is no choice but to reject how the Czech government as a whole has so far "resolved" the Roma issue since all previous decisions have been criticized not only by the general public but by academics as well, e.g., Albert (2020). This article presents a new paradigm, allowing a way out to find other strategies to address the Roma issue of children's education, strategies with lasting success that may bring hope for a better and more inclusive Czech society.



## 2. Migration Problems of Roma Ethnics from EU Point of View

Over the last five years, Romani citizens living in the Czech Republic have become more aware of the migration problems of their fellow citizens abroad. Families traveling from other European countries come back mainly from England, where they were unsuccessful in starting new lives, and some children of these returnees from abroad do not speak much Czech, or can only speak Roma and some Basic English because they went to English kindergartens. A Romani child is then at a big disadvantage when they attend primary school. Families have no money to properly support their children, and they usually live in a small apartment, where ten individuals frequently live in a single room. In a situation such as this, it is oftentimes more difficult for parents to monitor all of their children. Further, parents are under great psychological strain due to intermittent availability of hot water, electricity, and heat. There is also a very negative effect of scrapping housing supplements and the emergence of no-pay zones, which are demotivating, as previously mentioned in studies such as Vermeersch (2017). No-pay zones refer to areas in Czech-excluded localities, where the landlord rents apartments to Roma families, which are often in deplorable conditions and are often lacking in basic amenities (without access to water, electricity and gas) for a so-called contract rent, which does not correspond to the usual rent prices in the so-called normal, i.e., non-excluded locations. If a Roma family is late with their rent payment, the landlord turns off their water and electricity, the parents of Roma children are put under psychological pressure because they cannot cook a hot meal for their children, they do not have hot water for washing and they cannot warm themselves in the winter months. Of course, such behavior is illegal and contrary to the basic bill of rights, but there is not much willingness to do anything about it, and the state cannot legally order the owners of such properties to seek redress, as this is private property. In addition, many municipalities also act in segregation and against a sense of RSI. The isolated state of rented properties has a significant negative effect on children and youth. Families live in apartments that are small in size, and do not provide a proper environment for prepare Roma children for school.

Ciulinaru (2018) pointed out that the majority of people lack basic knowledge of the welfare system and often have no idea that moving to another location, even if it is within one municipality, can cost them extra money out of their already tight budgets. Some municipalities in the Czech Republic deal with a situation where there are localities (parts, streets, or just buildings) with an increased incidence of socially excluded phenomena on their territory. By these "phenomena", we mean the behavior of the population, which is characterized by, for example, an unhealthy lifestyle, violation of social norms and ethical values, the use of addictive substances, and a high level of indebtedness to the population. Of course, this argument is used by some speculators who buy real estate and rent it to these groups of residents for higher rent than the current market price—amounts between 5000–10,000 CZK (406 EUR) per month for rent (Public Decree Act, Statutory City of Ústí nad Labem 2018). Approximately 12 million Roma people live across the EU, and this is the largest national minority on the European continent. In particular, so-called socially excluded localities are regarded as a burning threat associated with Roma ethnicity. It would be best if they faced the truth. The price scale for human products and services has changed substantially, and ancient traditional Roma activities, such as wired or flying pots and grinding tools, are devoid of meaning. Inflation and the recent spike in the housing market are also absolutely not in harmony with the traditional Romani lifestyle. In contrast, Fraser (2009) points out a model situation that applies: Imagine that on a family unit level, a cracked or leaky pot nowadays is more likely to be thrown out and replaced by a new one, rather than repaired and this is similar to any agricultural, grinding or industrial tools on a company level. Seasonal agricultural work is short lived, and folklore and tradition as a way of income feed some percentage of the Roma population in terms of tourism. Expressed economically, the model is destined to fail as the supply is almost gigantic and the demand almost zero. These are the inherent problems of the current Roma population. Speaking about the commodities available to Romani people, if a few hundred citizens

travel to the edge of any western European city, they must also eat, they must buy petrol or diesel, and have access to basic amenities.

Bačáková (2011) informs that every state has different conditions, and there is a need to promote Roma education everywhere, not only in the northern Czech Republic undertaken by the Minister of Education, the Minister of Labor and Social Sciences and the government's human rights commissioner since Romani people are diachronically spread all over the Czech Republic, not localized in one particular area. Cahn (2017) argues that positive discrimination means, in particular, admitting to the majority population that this is their responsibility as a matter of priority; moreover, today's Roma people in the Czech Republic are predominantly immigrants from Slovakia and the majority of them speak Slovakian fluently, which is usually in contradiction to the belief that Czech Romani people ended up mostly in concentration camps of the Third Reich. So they, the current Czech Roma, have been uprooted twice, though they may not even actively know this themselves or do not identify as such, as previously described in historical studies such as Rochovská and Rusnáková (2018).

### 3. Segregation of Roma Children in Education

Messing (2017) says that according to the State of the Roma Minority in the Czech Republic, Roma children in education often encounter segregation. In the last school year, Czech Romani accounted for approximately 4% of pupils in the Czech Republic, and 15% of pupils in so-called special schools. O'Nions (2010) argue that nearly one-quarter of Roma boys and girls also went to primary school, where the majority of their classmates were Roma. Messing (2017) argues that the Czech Republic has long drawn criticism for its access to Roma children. Inclusion has several opponents in the Czech Republic. Human rights institutions call for states to fulfill their obligations and promote "belonging among children" through inclusive education, as Redondo describes (2011).

van de Bogaert (2019) points out that the UN, the OSCE and other organizations have today called on states to step up their efforts to include Roma children in mainstream schools. According to human rights institutions, Roma segregation in education is worsening, accas described by Rochovská and Rusnáková (2018). The Czech Representation of the United Nations (UN) has informed on this issue. Exactly a decade ago, the European Court of Human Rights condemned the Czech Republic for violating the right to education of and discriminating against Roman children by reassigning 18 of them to special schools.

The OECD reported (2018) that despite these efforts, segregation has not yet been eliminated. In fact, the institutions say that the issue has worsened. The call was jointly published by the UN Human Rights Office (OHCHR), the OSCE Office for Democratic Institutions and Human Rights (OSCE ODIHR), the EU Agency for Fundamental Rights (EUFRA), the European Network of National Institutions for Human Rights (ENNHRI) and Equines (European Network for Equality). Shmidt (2015), in his study, proved once again that children placed in segregated educational establishments are victims of discrimination and that the segregation model in education is not functional, but borderline abusive and criminal. According to this study, in separate schools, it is not only Roma children but also those with disabilities, those from immigrant families and those from mainstream society who are denied the benefits of co-education and models of diversity. Additionally, these children receive a substandard education that severely limits their opportunities to pursue all sorts of professions, which disadvantages them in comparison to their peers educated in other Czech primary and secondary schools. According to social organizations such as the EUFRA or the ENNHRI, children with poor education are doomed to take low-paying jobs and suffer from a life of social exclusion, as was also the conclusion of Schmidt's study back in 2015. According to human rights organizations, this verdict was ground-breaking. Strasbourg judges dealt with other cases of segregation in other European countries and focused their attention elsewhere. Obrovská (2018) investigated the decision of the Strasbourg court to make it illegal to treat Roma children differently under the pretext of their language problems. Concerning privacy and ethical procedures in Czech

sociological studies, the aim should be to also collect and publish data and statistics on minority education, which severely differs from the data provided by the Czech ministries. Roig et al. (2018) argue that we see the path to inclusion primarily in the introduction of a mandatory final year of preschool education in the form of nursery schools or prep classes for all children, as this was the focus of the education minister of the Czech Republic, which was reported to the OECD (2018). An alternative to a preschool focus might be the obligation to study an abbreviated type of study, such as a two-year course with a focus on practical and craft classes, as that is the contemporary practice in special secondary schools in the Czech Republic. It would extend the study period, but students would at least have the essential experience necessary for practice and applicability on the job market, rather than being led to be a common client at the Czech employment office.

### 4. Is There Any Point in Teaching the Romani Language in Czech Schools?

Eckert (2014) suggests that Czech schools should provide Romani language classes with the Romani language as an auxiliary language in schools in locations where children cannot speak Czech, a belief that was being put forward by conservative members of the parliament of the Czech Republic, as a means of helping them to overcome the language barrier. The Ministry of Education does not like the proposal of some members of parliament to educate Roma children. Segregated classes or teaching of Roma children in both Czech and Roma languages is considered counterproductive by the EU as well. Moreover, it violates the anti-discrimination law as well as the ruling by which the Court of Human Rights in Strasbourg committed the Czech Republic to the maximum inclusion of Roma children in education, as reported in a study by Cashman (2016).

Eckert (2016) points out that, in practice, it would look as if there would be special classes in schools for both Roma first- and second-graders, who would then switch to mainstream classes. For, according to them, Roma children no longer even understand Romani, says Samko (2019). Czech is our national language, and we see no reason to teach subjects in Romani—the experience is that some children do not speak Romani. Language is not a barrier when the baby is born here and has lived here for many years, says Eckert (2016).

In one of the schools, which is intended for children who have difficulty managing the demands of a mainstream primary school, the headmistress of that school is inclined to believe that Roma children cannot speak Romani, so there is no reason to teach some subjects in a unique language. In another of the schools, which is a primary school geared towards children from disadvantaged backgrounds, the experience is that children can speak both languages, both Czech and Romani. At home, the child and their parents speak Roma, but in school, they learn Czech. It is not a problem for them, explains the deputy headmistress. Moreover, she says, this is a problem for Roma children in prep school rather than first-timers, as Albert describes (2020).

So the question arises: Does it make sense to encourage the teaching of Romani to promote Roma culture? Children should learn about their culture at school. It might make sense at the level of elective subjects. However, we cannot imagine who would teach it. The question is, would Romani college-educated people tour schools to provide a few hours of instruction? Kyuchukov et al. (2015) argues that the second problem is that there are no Roma educators and so the children lack these role models. Roma representation is increasing in education. While there were indeed not many Roma in secondary schools a decade ago, there are Roma teaching assistants. But how many times do they only have a teaching certificate and are only being employed to make a point that there are Roma assistants in a school, ask Bačlija and Grabner (2014). It has not been possible to establish if there are any studies that conclude that it is better to speak only Czech to children at home or to make use of the bilingual approach of each of the parents, which may be the recommended opinion of several psychologists or experts. Moreover, how would this be achieved in a majority white school, where pupils of perhaps twenty different nationalities attend?

## 5. Inclusion in Education

Inclusion is just a word; it depends mostly on how we define it. Inclusion is often defined in terms of finance. During lessons, we can use support measures for all pupils, whether that is tutoring, interactive school aids such as whiteboards, electronic reading machines, tablets, and various mobile apps we have never been able to use before. Felder (2018) claims that inclusion is not just about money; it is about the skills of an educator. If, as a teacher, we find that a pupil excels in a particular subject or has other skills, then I do not need money to do that. If a pupil does not excel in study-important subjects, he can excel in other skills, in communication, in the humanities, in arts, in music, he can be manually skilled, and it is up to us, as part of inclusion, to recognize where to place a child, what to develop them in, support them, and advise them with where to go next, says Felder (2018).

Furthermore, cases where schools cut off Romani children at the start of fifth grade and label them as "stupid kids" in the transition to second grade are highly inappropriate. According to a study by Egi (2020), if we do that, then the child is carrying it along with them, leaving them demotivated already on a primary school level at the age of nine. Moreover, if these pupils decide to cut their primary school studies short, they are facing the danger of ending up at the unemployment office. So, we, as the Czech state responsible for our youth, let children aged 15leave school unsure of their lives only to find themselves in the employment office. When in school, Romani children are heroes sometimes for not having to finish all years of education, but once they cut their school period short, then they are on their own and have no claim for subsidies from the state.

Kostka (2015) argues that based on studies conducted, Roma is the most oppressed minority. The big problem in our education is ignorance of Roma culture and Roma in general. Kids encounter catchphrases such as "Hey Gypsy, he is bad...," but cannot explain why he is terrible afterwards. There is always a "but" so far. We as adults can understand the context, but children will take it differently and then condemn Roma as well, so our starting point is that not many of the majority children know anything about Roma culture, the history the Roma have been through in the 800 years they have lived here, why there is a distance between them and the majority. Children hear from adults that Roma do not work, says Kostka (2015). These are the kinds of myths that we need to debunk.

Qvortrup and Qvortrup (2017) claim that inclusion will only be useful if it stops being associated with Roma. Inclusion is not just for Roma children, but for all those with a handicap, a deficit, to have the right to the same education as children from the majority, healthy children. Inclusion is about every child leaving school with enough output to make the most of their potential for the future so that their education can be realized to the highest degree. So we ask, why not give it to them? School should be more tailored to children's needs, so we would envision such a change to see outright that we are developing a potential for children that remains hidden in today's educational conditions. If we know that "Novak" enjoys, for example, chemistry and that he would like to take it further, then educators should develop that talent of his within chemistry. Furthermore, on the other hand, I do not think it is necessary to give fives at all costs, to let a child fail in subjects they do not excel at knowing it can hurt them, says Thorjussen (2020).

Kohnstamm (2017) inform that some psychologists from educational-psychological counseling and teachers are still trying to suppress the use of Romani. "Gypsy parlance" still irritates. People such as that seem to have it rooted in them, and they cannot cope on their own if things are not going well for a child to learn how to spell OK—so the problem is that they speak "Gypsy/Romani language" at home. It always has to be blamed on something and the search for excuses continues. At the same time, there are a lot of educated Roma who, when privileged, teach their children from a very early age the importance of education and integration into the majority society; however, this is a difficult aspect as well since we do not know whether that that would be a handicap for them as a culture that is in pursuit of equality and integration to lose their own culture as well. On the contrary, it is essential that tradition, culture, and language are maintained. Should we also tell Vietnamese pupils (as a Vietnamese diaspora has been in the Czech Republic for

several generations now) who talk to each other in Vietnamese on their breaks to not talk so much because the teacher does not understand them? By now, we should certainly no longer consider a successful pupil one who, while thriving, is always perfectly prepared for the classroom, is obedient, but does not express his opinion, merely accepts knowledge and reproduces it perfectly. Such an individual carries signs of mediocrity, even though from a behavioral and learning perspective they are among the so-called "perfect" pupils. Forlin and Chambers (2020) argue that school success is certainly not about good grades and perfect behavior, as only a great memory and mechanical learning capability are insufficient for the world today. The kind of pupil who asks questions, thinks about the problem, looks for his own solutions should be considered successful. He will not settle for just saying that 'it is so'. He wants to know more. Byrne (2020) claims that thoughtful and inquisitive pupils are often very restless and their behavior cannot be described as perfect. We can also rate as successful a pupil who, while not very prompt, cannot find new avenues on his own, is interested and keen to participate in the life of the school and class and its activities, collaborates with classmates and through his work aids a good outcome within the team, says Brown (2020).

According to Haug (2019), pupils with fully developed skills, creativity and teaching commitment cannot be assumed on entering the classroom. Developing these factors is the job of teachers. The school should help pupils discover their area of interest, build personal commitment skills in the field and work creatively in it. For every pupil has a chance to reach their personal best and be successful at school, they have to have space. A teacher should help all pupils, based on their abilities, to gain new knowledge and skills and thereby develop their personality. In reality, however, the situation in some schools looks different. Many pupils experience success very rarely or not at all, and this is certainly not and cannot be an incentive for them to improve school performance or increase interest in further study. Such pupils, for whom success in school is often an unfamiliar term, include most Roma children, says Haug (2019). In most cases, Roma children copy the unsuccessful educational paths of their parents. Increasing school success, which would result in the acquisition of qualification and, together with it, the ability to apply to the labor market, is, therefore, a clear priority in the field of combating the social exclusion of Roma and is one of the main factors influencing the chances of Roma pupils to successfully complete primary school and subsequently obtain a qualification.

## 6. Multicultural and Intercultural Education in the Czech Republic

Takeuchi (2018) thinks that multicultural education arose as a trend when people became aware of conflicts in cultural contacts. This situation occurred in states that can be described as immigrant states and confederacies such as the USA, Canada, and the Netherlands. Some of these states have emerged as immigrant states because they have been affected historically by waves of immigration (USA and Canada). Other countries have initially been rather homogeneous culturally but became the target countries of immigrants (e.g., the Netherlands within the EU states). The awareness of multiculturalism, as well as multicultural education, continued to spread to the UK and Australia. In the 1970s, the ideas of multiculturalism spread throughout the world.

Until 1989, the Czech Republic was a closed country that had not addressed the issue of multiculturalism or ethnic coexistence in the recent past, says Marincova (2018). Most of the population of our country was not interested in other national or ethnic groups, their history, value patterns, and behavior. In the Czech Republic, only 5% of the population is currently a national minority. However, this number will increase by the increase in certain minorities and also by the fact that people from different countries are coming to work for us and probably in the future because our country is part of an open Europe. Each national or ethnic group has its history, culture and value patterns of behavior. The cooperation of several nations (ethnicities) can lead to misunderstandings and conflicts, as each group (national and ethnic) has different values, customs, and cultures. As with prejudices, stereotypes stem from the same psychology, in which rational content is mostly

suppressed, and at the same time, they have a strong emotional charge. These are the attitudes, opinions and ideas that an individual or group has toward another individual, group, or towards himself. This can be national, ethnic, racial or religious stereotypes. They are template-like ways of assessing and perceiving individuals, groups, classes of individuals or objects, which are not based on the individual's direct experience but are adopted and maintained by tradition (Průcha 2010). Modood and May (2019) inform that people of different nationalities or ethnicities living in the same territory must know each other well, understand and do everything for a positive coexistence. Every group can enrich the life of society as a whole. The maturity of the majority of society is also evidenced by the quality of relations and coexistence with minorities, as previously described by Uherek (2018).

Hidalgo-Capitán et al. (2019) presents an opinion that the aim of multicultural education is therefore to create a civil society in which people will respect, tolerate and culturally enrich each other. The term "multicultural education", as it was introduced in Czech pedagogy from international terminology in the early 1990s, is mostly used in Czech professional terminology. At present, this term is more or less common in other sciences (ethnology and sociology) and the field of educational practice. In addition to the term "multicultural education", there are sometimes terminological variants: "intercultural education" and "intercultural education." Multicultural education is a preparation for the social, political and economic reality that pupils experience in culturally different human relationships, says Arsal (2019).

The term multicultural education expresses efforts to create, through various educational programs, the ability of people to understand and respect cultures other than their own. It is of great practical importance concerning developing attitudes towards immigrants, other nations, and races. Intercultural education promotes tolerance and the ability to understand each other and rejects ethnocentric thinking. Lee (2013) informs that intercultural education emphasizes the importance of the mother tongue for the psychosocial and cognitive development of immigrant children and promotes bilingual education. Intercultural education can only be implemented successfully if school, family and society work together. Even the Czech Educational system has the potential fo fulfil the concepts of multicultural education (see Figures 1 and 2. below) as the last Czech Ministry of Education report (Czech Ministry of Education, Youth and Sports 2021). The concepts of multicultural and intercultural education often overlap. According to Guild (1994), there is little doubt that there is a link between the culture in which pupils live and their preferred way of learning. Additionally, this relationship is directly related to academic, social, and emotional success in school. Durodoye and Hildreth (1995) point out that although the relationship between culture and learning style exists, it is not yet fully explained. However, cross-cultural differences in learning styles are documented by a wealth of research. According to Guild (1994), this connection between the culture in which students live and their preferred way of learning, as previously stated, directly relates to academic, social and emotional success in school. It may happen that the cultural and educational traditions created by the relatively homogeneous majority of the population of a given state differ from the cultural and educational traditions of the minorities in the given state. Indeed, children from various minorities have problems with their learning styles from the beginning of their schooling. Educators have noticed this fact in many countries of the world. One of the approaches in the framework of multicultural education is the effort to increase the school success of selected groups by means of changing the curriculum, which would respect cultural peculiarities. In her study of culturally specific curricula, Dunn (1997) found that whether students were able to learn certain content was dependent on the way they learned it, not the content itself. Thus, they argue that a culturally sensitive curriculum cannot improve the school performance of disadvantaged cultural groups. From the above overview of current findings on the investigated problem, it can be seen that many studies deal with the issue of intercultural differences in learning styles and their effects on school success. Much of this research also focuses on comparing the learning styles of ethnic and

cultural minorities with the majority population of a given country. However, these are mainly foreign, especially American, studies dealing with significant minorities in their country, e.g., African Americans, ethnic Indians, and Mexicans.

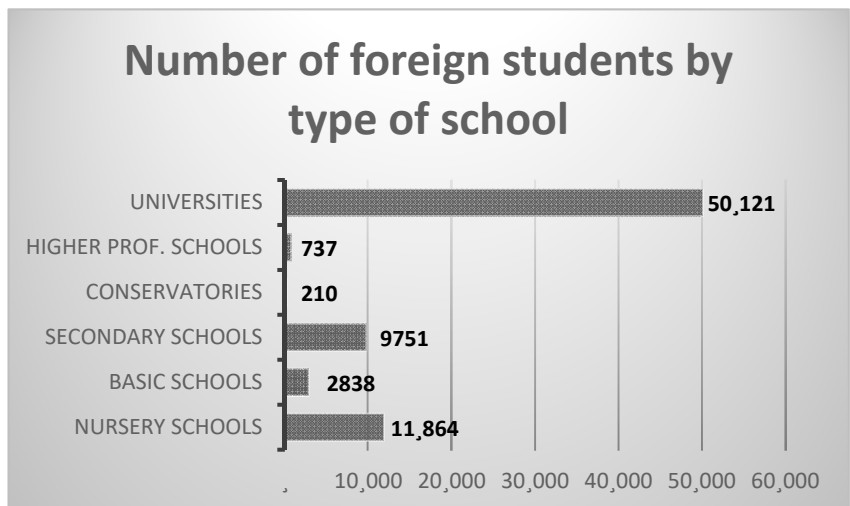

**Figure 1.** The number of foreign students in the Czech educational system in 2020/2021 (Czech Ministry of Education, Youth and Sports 2021).

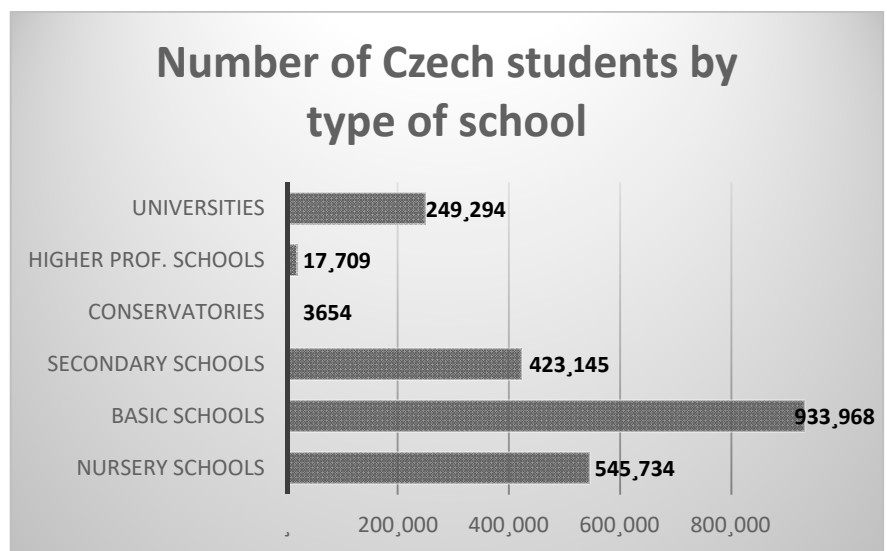

**Figure 2.** The number of Czech students in the Czech educational system in 2020/2021 (Czech Ministry of Education, Youth and Sports 2021).

Rýdl (n.d.) informs that in Czech education, multicultural (intercultural) education began to appear only after 1989. The reason was a new social situation when various sociocultural groups started to speak, and the number of foreigners increased. The Czech education system began to change only slowly. International and non-governmental organizations have played an essential role in this period. In early 2001, the People in Need Project Variants project started. Within the project, information about minority groups was disseminated. However, the main objective was to introduce new principles into the education system focusing on the skills and attitudes of pupils and students, says Grant (2018). First of all, the project implementers tried to encourage teachers to apply intercultural education (IKV) in their teaching. Within the project, we managed to prepare a chapter of the Framework Program for Basic Education, which should gradually become part of the new education legislation. A handbook "Intercultural education not

only for secondary school teachers" was published. The Variants project also concerned the analysis of textbooks for primary and secondary schools, which gave rise to several recommendations for the approval process through which all textbooks go through. The variations encouraged every textbook that receives a Ministry of Education endorsement to meet the criteria for multicultural tolerance, says Makarova (2019).

The foundation of multicultural education is the creation of a multicultural community in nursery schools, classrooms and the school as a whole. All children, regardless of nationality, ethnicity, religion, social or other groups, should gain the experience of coexistence, which is characterized by equality, cooperation, helpfulness and justice, says Karousiou et al. (2018). Minority children should feel comfortable in school facilities and broader society, and the school should contribute as much as possible. Multicultural reciprocity is also an essential experience for the majority of children. Thanks to this type of education, children and young people can prepare for life in an environment where they will meet members of other nationalities, religions, races and ethnicities. The impact on pupils in the field of multicultural education is not only relevant to educational content, which is presented in educational programs and textbooks, but also to the climate of the classroom and school against the broader social environment (country), says Senyshyn (2018).

Multicultural education is not a specific subject or subject matter, and it is a pervasive aspect that permeates all pedagogical efforts (teaching, education, and leisure). Pupils encounter the topics of multicultural education in Czech language and literature, history, and geography, which have a close connection with multicultural education. Teaching attitudes, value systems and the behavior of pupils are corrected. Multicultural education is part of the educational offer of the school educational program, which is created based on the requirements of the Framework Educational Program for Preschool Education, says Slavkovic and Memisevic (2019).

### 7. So Why Are Roma Pupils Unsuccessful and Can It Be Changed?

The school failure of Roma pupils mostly stems from their lack of preparedness for school life. The causes of this unpreparedness lie in a different way of raising children in the Roma family and the significant lack of interest of Roma parents in preschool education. Small Roma are thus unable to acquire necessary language skills and the lack of mastery of the Czech language then becomes a significant problem in the first contact with the school. As a result, there may be neuroses and, of course, inappropriate behavior during classes. Another reason may be a lack of knowledge of the way of life in the Roma community by teachers.

Kóňopvá (2019) in her work "Approaches of Roma parents to the education of their children" states that Roma parents show interest in their children's education in most cases only in the first years of compulsory schooling (usually up to the 3rd grade). In practice, their further lack of interest manifests itself in the fact that the children gradually experience educational problems. At this moment, the interest of Roma parents in the education of their children is gradually declining. They then shift the responsibility for educational and educational problems to the children's pedagogues. They expect that teachers will be the ones to solve problems for them, that they will be able to solve even those problems that arose in the family environment and may not even be related to school (Kaleja 2009). At home, Roma parents often do not ask about their children's homework and often do not even check notebooks or schoolbooks. They, therefore, do not have an overview of how the child is progressing in school or does not benefit. These parents are not aware of the educational "participation" in their children's education. Roma parents often live in the idea that the education of their children is fully within the competence of the school and that the school should show the necessary efforts and efforts towards their child in order to be able to support him in his development. In practice, we can notice that Roma parents or grandparents (mostly grandmothers) accompany their children to school in the first years of compulsory school attendance. Some wait for the children on the school premises

until classes start. There are also parents who ask about the child's schoolwork, and some consult with the teacher about homework (not so common).

On the other hand, a comparative study by researcher Anita van der Hulst (van der Hulst 2020) gives hope that a better situation can be traced in some aspects of the ratio of families to secondary school and university studies of Roma children. In the Czech Republic, there are more parents who encourage their children to achieve a higher level of education, because they realize that education is a necessary means of social advancement (van der Hulst 2020, p. 63). After attaining higher education, a paradox appears that we could call "reverse stigmatization". The point is that a Romani man or a Romani woman who, with the great support of his family, mainly his mother, has achieved secondary or higher education and managed to get out of socioeconomic poverty, is considered a "gajo" (white foreigner) in the Roma community, i.e., "Non-Roma". Highly educated Roma are trying to get rid of the stigma, as reported in an interview with a Czech Roma woman, Marie (born 1990). Therefore, they prefer not to draw attention to their Roma origin in the middle-class environment in which they live. Additionally, that is precisely the problem, Marie thinks (van der Hulst 2020, p. 60). Thus, they become "invisible" both to statisticians and policy makers, but also to other Roma, to whom they should be the image of a socially successful person. Nevertheless, the research discovers "pioneers" who want to "consciously create their own middle class and see themselves as an example for their own environment (...) Only as a Roma middle class can they function as a role model for the "gajos", but especially for Roma and Sinti with low education and lack of chances." (van der Hulst 2020, p. 63).

Cashman (2016) states that for the education of Roma pupils, a teacher should acquire special knowledge and skills. It should address, among other things, the characteristic expressions of Roma personality. This type of Czech perceived personality can be defined as a person who considers himself/herself to be Roma, without necessarily claiming this affiliation under all circumstances, and/or is considered such by a significant part of his/her environment on the basis of real or imagined (anthropological, cultural or social) indicators. Previously, the designation Gypsy was not used for an ethnic group, identified by skin color, but for persons living a certain way of life. A significant characteristic of individual Roma sub-ethnic groups was a specific profession, which was passed down from father to son for countless generations. In traditional society, intergroup social distance manifests itself in endogamy—members of one sub-ethnic group are allowed to marry and marry exclusively among themselves, and other marriages are often inadmissible or significantly restricted by complex rules.

For in order to better understand some of their acts and behaviors, we must first recognize the differences of Roma ethnicity.

According to Gilham and Fürstenau (2019), Roma is oriented only toward the present, so it makes little sense to try to arouse their interest in further study by creating ideas about future occupations. We hardly find a single publication dealing with Roma issues that do not emphasize the fundamental importance of education for the rise of the Roma ethnic group and its integration into society. In the same way, they all agree that Roma children are very unsuccessful at school, that the approach of Roma to education is not equal, and also that the current education system is not prepared for the cultural specificities of students and for a differentiated approach to them. In recent years, we have published a number of publications that deal with these questions in great detail.

The level of education of the Roma population is currently very low. In the older age groups, there is a relatively large representation of illiterate people, and the younger age groups are often semi-literate, mostly having only basic education without further qualifications. The situation of the youngest generation of Roma is also unsatisfactory. Research has shown that, compared to other children, Roma children fail 14 times more often, are classified as second or third grade behaviorally 5 times more often, drop out of school 30 times more often in the final grade, and are transferred to special schools 28 times more often (Šišková 1993). These facts have adverse consequences in the social,

economic and cultural spheres; they deepen group and social contradictions and cause significant problems in terms of national understanding. According to Šotolová (2001), the condition for increasing the educational level of the Roma ethnic group is the acceptance of ethnic, social and cultural differences in the educational process. However, current experience with the education of Roma children indicates that our schools for the time being do not respect the ethnocultural, social, linguistic and psychological differences of Roma students and the differences in their life perspectives and goals. According to Holomek (1997), the current primary school enhances rather than dampens the ethnic differences and handicaps Roma children. Šotolová (2001) considers our school system to be quite inflexible, because the educational process primarily counts on children equipped with such personality and family dispositions that ensure a conflict-free existence in school. Therefore, if a child deviates from the normal, he is not always provided with conditions for obtaining an education that would correspond to his abilities and possibilities. Other European countries also report on similar experiences with the education of Roma children. For example, Chronaki (2005) states that the participation of Roma children in schools in Greece is unsystematic, and emotionally traumatic and can also be characterized as an experience of constant cultural conflicts. Tauber (2003) also documents a similar experience in her observation of the school attendance of children from the Sinti Estraixaria group—the Roma community in northern Italy. He states that the Italian school is only very poorly prepared for the education of Roma children. Greater emotionality and little resilience must be taken into account when preparing educational activities for Roma children. Most Roma accept the societal rules of the majoritarian society only mechanically, showing significant volatility, and this is then reflected in their hierarchy of values, leading to a different conception of morality. Problems in coexistence between Roma and the majority are indeed increasing in some cities, but they see the blame on the side of the state and municipalities. According to him, the solution is in the hands of the Roma themselves, who must gain self-confidence, trust in their own abilities and the will to get out of the bottom of society. The founder of the civic association Vzájemné soušítí, a native of the Indian state of Kerala, originally a teacher, came to the Czech Republic after the floods in 1997 to help the suffocated Roma. He had already stayed in the city and gained respect as an expert on Roma issues, enjoying the trust of both Roma and the Czech authorities. For many years, Roma lived in larger numbers in some parts of the Czech Republic. There have been anti-Roma sentiments before, but there have never been sharp conflicts. Now, on the one hand, there is momentum from the outside—extremists are trying to gain a majority against the Roma, and certain forces suspect that the road to power leads this way. Additionally, then—mainly due to developer interests—we see efforts to clear some localities of Roma. So the question is: *What are the most common clichés about the Roma?*

Above all, the Roma are better off than the whites. People say: "I work for a stupid 20 thousand, a gipsy doesn't work and gets the same money from the state." But he no longer sees that 15 out of the twenty thousand goes to the owner of the hostel for miserable housing. I know Roma who are hungry; they have no food. The law applies equally to all. it is not as if a gipsy comes to the office and if shouts a lot, he gets 10 thousand more in benefits. A civil servant who would function in this way would lose her job. This idea is exactly the type of folklore that spreads unchecked. But we must try to keep our thoughts objective, and not fall into hysteria and barbarism. As long as the Roma do not believe that they can rise from the bottom, they will not attempt anything.

A Roma guy who goes and steals earrings from a white girl's ears is a person who does not respect himself. Anyone with a little self-respect would not behave in that way. So we try to strengthen their pride and self-confidence. That is why, for example, every Friday, Roma women meet and sing Roma songs together. It is about them hearing their common voice, and learning to sing fully, and proudly. We hold a meeting of people from the Roma community, where everyone can raise objections, and discuss problems. In order to feel togetherness, they learned to overcome differences and solve neighborhood problems. The Roma must unite, gain pride, and find a common voice. When this is achieved, we can

talk about establishing Roma assemblies after Ostrava, a kind of Roma self-government. Then, the Roma will be able to come and say: "Give us a job". The way it happened in Liščina, where we made an agreement with an investor and employed people to repair houses. Without emancipation within the Roma community, nothing will happen by itself. Additionally, emancipation is joint decision making, the ability to bear responsibility.

Marliani (2020) thinks that the different history and the related way of life of today's Roma also play a significant role here. Roma parents raise their children in a different way to Czech parents. In the misunderstanding of these differences, there may be causes of school failure for Roma pupils, as well as causes of negative relationships with schools and a distaste for education.

Marliani (2020) argues that, above all, any measures put in place at the school must be supported by societal changes that will contribute to greater alignment of the majority's coexistence with the Roma ethnic group. It is no secret that in our society, there are negative relationships between Roma and the majority population. Peček and Munda (2015) inform that systemic changes in education should be directed towards the individual integration of Roma pupils into mainstream schools. Teachers must internally adopt the strategy the school will develop to increase the success of Roma pupils. Payne (2019) informs that educators should know the history of Roma and their traditions. Knowledge of the Roma language, even partially, would also make a significant contribution to positively addressing the problem of school success in schools, says Szilágyi (2015). A teacher should use all appropriate support measures and apply methods and forms of work that would pique not only the interest of Roma pupils but also change their parents' relationship with the school. Relations between teachers and Roma parents should be built on trust and genuineness. Parents' positive attitude towards school is essential, says Marcantonio (2019).

One of the foundations of school success, especially for younger children, is to be able to give them a learning substance in the form of play that replicates their interests. A teacher should recognize the personality of each pupil because only then can they create activities that will direct towards the individual development of children. The benefit does not show the image of a child's personality objectively. His real personality, which should be further developed in school, only becomes apparent during playing games. Samko (2019) points out that, based on the studies conducted, dealing with the Roma issue, it can be directed that when we compare the teaching practice of "normal" schools and schools in excluded localities, which, moreover, are increasing all the time, it is impossible to apply only what is shared in a majoritarian society. The bilingual school, where pupils of perhaps twenty-two different nationalities attend, is an example of this. Bilingual education has a long tradition, e.g., in Luxembourg, it has existed since 1843. In the 1970s and 1980s, there was a Canadian experiment with so-called immersion language programs, which were supposed to ensure the teaching of French and French realities in schools. It was the Canadian immersion that had a great influence on some types of European bilingual education programs (García et al. 2008). The concept of language immersion is divided according to the amount of foreign language teaching and the level of education in which immersion education begins. Total immersion education means that the entire educational content is taught in a foreign language from the beginning of the student's entry into formal education. Partial immersion means that certain parts of the educational content are taught in a foreign language. We can also talk about the so-called early immersion, which is implemented from kindergarten or from the 1st year of elementary school. Late immersion starts in the 2nd grade of elementary school (Coyle et al. 2010).

Bilingual education in the Czech Republic is implemented in primary schools on the basis of the Instruction of the Minister of Education, Youth and Sports on the Procedure for Permitting the Teaching of Certain Subjects in a Foreign Language dated July 15, 2008. For bilingual (=bilingual) grammar schools, a trial period is taking place between 2009 and 2015 verification of teaching according to the pilot Framework Educational Program for bilingual grammar schools, which takes into account the specifics of these schools. The CLIL method and other forms of language immersion and bilingual teaching have many

common features. CLIL is sometimes referred to as one approach to bilingual education (Graddol 2006). Bilingual teaching leads to mastering the language almost at the level of a native speaker and focuses mainly on receptive skills. The CLIL method focuses on both receptive and productive skills. The development of cognitive skills and thinking in a foreign language are the key benefits of this method.

In the field of education, the integration of the socioculturally disadvantaged (e.g., Roma) and the medically disadvantaged is most often discussed. Integration does not have to concern only the Roma group. However, a group or an individual is always integrated, significantly differently from what is in the majority. The group is considered standard. The goal is to make this group competitive, and equal, ensure qualification and, last but not least, improve living and social standards. Roma children and their integration into the non-Roma majority have been a topical and debated topic since 2004, when the Czech Republic became a member of the European Union. However, this issue cannot be addressed without the parties involved having a historical and cultural awareness of the Roma community as a whole. Thanks to great media coverage, Roma children currently have more possibilities, and more and more doors are opened for them, whether that is in terms of education, interest groups or organizations providing free time activities. As we all know, there are two sides to a coin. It is a question of what percentage of integrated Roma children will retain their goals, ambitions and vitality to continue their studies at a higher level even after graduating from primary school. It is not unusual for a Roma child to leave primary school with huge plans for the future, without any problems passing the entrance exams to secondary school, where he will also enter. He maintains his position for a while, and then later he begins to "falter" a little in learning, which is usually the beginning of truancy. Gradually, the parent does not make them complete the missing curriculum and thus deepens the gap that is slowly forming between the child and the school.

Samko (2019) further claims that Roma parents are sometimes reluctant to let children go on trips, especially girls, who have a hard time when they fill the role of helper and nanny in the family, and this is hard with Roma because girls are perceived differently—even a 12-year-old girl will already be at home doing the work of an adult woman: cooking and taking care of younger siblings, (Bullivant 2019). It is suitable for children who live a life confined to an excluded location to open up their horizons, motivate them so that their view of how to live life changes so that they perceive it differently. Bullivant (2019) claims that taking them on a trip, to the seaside, showing them that they are not just expected to go home or be at home, cooking and looking after their siblings, but that they could travel, educating themselves so that they might have a better job, showing them where that life might lead them. Humphris (2019) says, Moreover, we should not chop them off at the outset and let them know that they are useless in society because if we do that, those 12- or 14-year-old girls will then one day become mothers just as easily as their parents. Barely a well-behaved child will raise themself, and that is wrong, at least according to the findings of Cintulová and Radková (2019), who focused on Romani youth and young girls within the Roma/Romani community in the Czech Republic.

## 8. Conclusions

An individual education plan should be a suitable solution for each integrated pupil no matter the cultural background. However, this plan must be an open document which can be changed according to the development and progress of the pupil during the school year. The unpreparedness of Roma children for school conditions is what schools are trying to address by setting up prep classes in the Czech Republic. Graduating from the prep year is voluntary, with interest slowly rising, according to statistical investigations, but unfortunately, the initial enthusiasm of some Roma pupils often fades over the school year due to lack of motivation. Irregular attendance is again causing these pupils not to develop the desired habits to a sufficient degree. One option would be to introduce compulsory attendance in the final year of Czech pre-school programs. Such a measure would benefit not only Roma children but all preschool-aged children who have never

attended kindergarten. A preschool program for a Roma child will help to compensate for differences in language, develop motor skills and lead to the creation of not only social but also essential hygiene habits for some children, which is a prerequisite for more accessible adaptation to the school environment. Out of all the mentioned studies, the conclusions are common in suggesting that if the school success of Roma pupils and their interest in the further study are to increase, it is not enough to introduce the necessary measures in schools, it must be a whole-of-societal change that will hit far more areas of ordinary life. How might clinicians/practitioners in the Czech Republic support the education of Roma children? The specifics of the education of Roma children and pupils expand the theoretical base by offering specific activities and activities that can be used in the education of Roma children and pupils in pedagogical practice during effective communication with the Roma pupil and parent. It also focuses on the field of career counseling as one of the tools to support the student during the transition period. The educational course will present proven tools, methods, techniques and activities that contribute to the development of key competencies and literacy skills that students need to apply in life. Through selected aspects of the education of Roma pupils and students, participants will gain a comprehensive idea of the traditional Roma family, the specifics of the education of Roma children, educational resources and forms of learning in the context of interpersonal relationships influenced by collectivism. Participants will learn about the peculiarities of the Roma ethnolect of the Czech language as applied to the school environment. The educational content focuses on the issues and factors of social exclusion, as well as the problems and impacts of early school leaving. Participants will receive an overview of the possibilities of supporting Roma pupils and students in the field of education, including information on current subsidy support. So what are the implications for policy? What specific recommendations would you make to policy in the Czech Republic to facilitate the education of more Roma children? The analysis of attitudes and educational needs pointed to the low chances of Roma children staying in the primary school classes they enter. The survey carried out in selected years of nine primary and special schools shows that out of ten Roma children who enter primary school in the first year, only three boys and half of the girls remain in their original class. The rest either fail at least once or go to a special school or class. Approximately one-third of the Roma pupils we examined, who started their educational career in a regular primary school class, ended up in these classes. In the majority population, approximately one child out of ten leaves the original class. Another big problem for Roma children is the significantly higher rate of missed lessons than for children from the majority population. Absenteeism rates continue to increase over the course of education. Statistical analysis has shown the effect of absence on pupils' performance. However, the higher absenteeism of Roma children is not only their reaction to poorer performance. The fact that Roma children are absent more than children from the majority population also applies to groups of children with the same grades.

**Author Contributions:** Study conception and design: M.M., data collection: L.F.R.; analysis and interpretation of results: M.F., M.M.; draft manuscript preparation: L.F.R. All authors have read and agreed to the published version of the manuscript.

**Funding:** Supported by the HORIZON2020—AMASS/ID 870621.

**Institutional Review Board Statement:** Not applicable.

**Informed Consent Statement:** Not applicable.

**Data Availability Statement:** Open access license.

**Acknowledgments:** Supported by the HORIZON2020—AMASS/ID 870621, Grant agreement ID: 870621. Call: H2020-SC6-TRANSFORMATIONS-2019. Open Access.

**Conflicts of Interest:** The authors declare no conflict of interest.

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
