# Peer review of "Social Services Management in the Context of Ethnic Roma Issues in the Czech Republic with a Focus on Education for Roma Children"

_socsci, doi:10.3390/socsci11100477_

Round 1

Reviewer 1 Report

It literally jumps from one issue to another. Some mistakes are so basic: Sicic at all instead of Siciv et al.  Line 26 ....says Urh   Line 28 Says Albert......

Line 33: Over the last five years, they have been more aware.... Who are they? The title is misleading for this section because it gives the impression that the section is about the policies and views of the EU. Instead, the paragraph is about returning Roma children from the UK, and their experience in not finding hot and cold water. 

One can endlessly comment on each line of the paper. The paper needs a total rewrite - my suggestion is to make it clear that you are focusing on Roma children, and the role of education for better reintegration. 

The paper should also be corrected by a professional proofreader - too many awkward sentence construction and mistakes. 

Some suggestions: Make sure that the abstract gives a very clear idea of what the work is all about: the research questions/themes you plan to discuss, the methodology employed, and the reached conclusions. The introduction should give an overview, and capture the reader's interest. 

Most importantly - do not jump from one theme to the other - but write a smooth 'story' that is pleasant to read and is focused. 

In general, as a paper, it summarises what many writers think about this and that, but not really what the authors think, and when this is provided, it is based on poor argumentation, without any data to back it up. 

Author Response

Line 33 has been corrected - they referring to the Romani children

sic et al corrected

data added

Reviewer 2 Report

Please check once more if all the authors mentioned in the text are listed in the references. There are authors which are in the text but not in the list of reference. And vice versa there are names of authors in the references which are not mentioned in the text. Please check!

The article is a very informative one. It does not have an empirical or experimental part, but the structure of the article is done in such a way that gives a lot of information about the topic of Roma education. In psychology these kind of article are called meta analyses of a particular issue. The paper is well written and easy to read. 

Author Response

thank you for your feedback

the non cited references are in general Czech governmental documents and census data and local studies

Reviewer 3 Report

Social Sciences

Review Letter to Authors

Manuscript Title:           Social services management in the context of ethnic                                               Roma issues in the Czech Republic with a focus on                                                education for Roma children

Manuscript ID:             Not known

Date:                              Sunday, July 3, 2022

_________________________________________________________

This study examined the substantial number of Roma individuals that have entered the Czech Republic, and how many Roma children are not receiving an education.

While I found the topic interesting, it was difficult for me to understand the author’s main premise. The author made many statements that assumed the reader was familiar with the Czech Republic and Roma peoples and did not provide further clarification of these statements. It is in the spirit of strengthening the manuscript that I offer the following questions/comments/recommendations:

On Page 1, you wrote:

Abstract: As many as 350,000 ethnic Roma live in the Czech Republic, according to qualified estimates. Yet only a fraction of the Roma population has received higher education and are achieving careers within and not alongside the majority society. Better integration, some believe, is meant to facilitate long-term community work and the vigorous removal of prejudice. It's quite a force that

could appreciably affect the political landscape. Unfortunately, we run into the disunity of the Roma themselves. This should change for the future. The current government concept for this purpose supports the development of Roma cultural and educational life.

·        This Abstract is too vague. Who provided these “qualified estimates?” Instead of saying “a fraction of the Roma population has received higher education and are achieving careers,” provide specific numbers here. Of course, provide a citation or citations regarding where you received those numbers. Who believes “better integration” will help facilitate long-term community work and the vigorous removal of prejudice?” Instead of writing “it’s,” you should write, “it is.” Unless you are using a direct quote from a source, you should not use contractions in academic writing.

On Page 1, you wrote:

Labour rejects the claim that those members of Czech society who object to the current arrangements and so-called solutions to the Roma issue are racists.

Change to:

Labour rejects claim that those members of Czech society who object to the current arrangements and so-called solutions to the Roma issue are racists.

On Page 1, you wrote:

Over the last five years, they have been more aware of migration problems.

·        Who is the “they” to which you are referring? Be clear.

On Page 1, you wrote:

A kid like that, when he goes to school, he has a hard time. Families have no money. They live in one apartment where even ten are in one room; then it’s harder, to keep an eye on all the children and attend to them as if there were five of them living there. Besides, parents are under great psychological strain due to switching off the hot and cold water, electricity, heat. There is also a very negative effect of scrapping housing supplements and the emergence of no-pay zones, which are demotivating, point Vermeersch (2017)

Change to:

When a child has attended English Kindergartens, he oftentimes has a difficult time when she goes to school. Generally, his family has no money, and he and his family life in a small apartment, where ten individuals frequently to live in a single room. In a situation like this, it is oftentimes more difficult for parents to monitor all of the children. Besides, parents are under great psychological strain due to switching off the hot and cold water, electricity, and heat. There is also a very negative effect of scrapping housing supplements and the emergence of no-pay zones, which are demotivating, point Vermeersch (2017)

·        What do you mean by “scrapping housing supplements?” What is a “no-pay zone?” In addition, did you mean to write, “which are demoralizing?”

On Page 1, you wrote:

Ciulinaru (2018) pointed out that people lack basic knowledge of the welfare system and often have no idea that moving to another location, even if it is within the locality, can cost them extra.

·        What do you mean by “extra?” Mention the specific monetary amount (and provide a citation).

On Page 5, you wrote:

Many pupils experience success very rarely or not at all, and this is certainly not and cannot be an incentive for them to improve school performance or increase interest in further study. Such pupils, for whom success in school is often an unfamiliar term, include most Roma children, says Haug (2019).

·        How is success defined? Be clear.

On Page 5, you wrote:

The cohabitation of several nations (ethnicities) can lead to misunderstandings, misunderstandings and conflicts, as each group (national, ethnic) has different values, customs, and culture.

Did you mean to write?

The cooperation of several nations (ethnicities) can lead to misunderstandings, misunderstandings and conflicts, as each group (national, ethnic) has different values, customs, and culture.

On Page 6, you wrote:

The concepts of multicultural and intercultural education often overlap.

·        You should provide a graph that outlines the difference between these two terms and where they overlap.

On Page 7, you wrote:

According to Gilham and Furstenau (2019), Roma is oriented only towards the present, so it makes little sense to try to arouse their interest in further study by creating ideas about future occupations.

·        As written, it seems like the author does not feel there is any value to teaching the Roman people about future occupations because they are only oriented toward the present. You seem to be generalizing here and you cannot generalize ALL Roma people. Even if what you wrote here were generally true, how would a multicultural lens help the Roma people expand their view of the present to encompass the future? 

On Page 7, you wrote:

Most Roma accepts the societal rules of the majoritarian society only mechanically, showing significant volatility, and this is then reflected in their hierarchy of values, leading to a different conception of morality.

Provide an example to support this statement.

On Page 7, you wrote:

So why are Roma pupils unsuccessful and can it be changed?

The school failure of Roma pupils mostly stems from their lack of preparedness for

school life. The causes of this unpreparedness lie in a different way of raising children in the Roma family and the significant lack of interest of Roma parents in pre-school education. Small Roma is thus unable to acquire necessary language skills and the lack of mastery of the Czech language then becomes a significant problem in the first contact with the school. As a result, there may be neuroses and, of course, inappropriate behaviours.

·        What do you mean by “neuroses” and “inappropriate behaviors?”

On Page 7, you wrote:

·        It should address, among other things, the characteristic expressions of Roma personality.

·        What are “the characteristic expressions of Roman personality?” Be clear.

On Page 7, you wrote:

The Billing school, where pupils of perhaps twenty-two different nationalities attend, is an example of this.

·        Say more about The Billing School. Where is it located? How long has it been established? What is their learning mission? What is their teaching pedagogy? How, specifically is the teaching pedagogy at The Billing School support your focus on Roma Children?

Implications for Clinicians

How might clinicians/practitioners in the Czech Republic support the education of Roma children?     

Implications for Policy

What specific recommendations would you make to policy in the Czech Republic to facilitate the education of more Roma children?

OTHER ISSUES:

My greatest issue with the manuscript is that the author does not make it clear from the onset [in the Introduction] what she is examining, why it is important, and why the reader should care about the topic. The author must provide specific statistics [based on trends over time] AND the negative consequences of not receiving an education.

You mention early education [Kindergarten] and higher education [college]. Which is your focus and why? Which Roma children are generally not receiving an education? Males? Females? Both? What is the negative consequence for themselves, families, and communities when they do not receive an education? What is the positive consequence for themselves, families, and communities when they do receive an education?

Are Roma people excluding themselves because they see integrating with the Czech Republic a dilution of their values as a people?

What specific things do you recommend to help people of the Czech Republic embrace the differences between themselves and Roma people (children)? What difficulties are present? What opportunities for change are present?

What kind of careers do Roma people generally have? Are these careers based on gender or ability to do the work? What is the required education level for these careers?

On Page 1, you wrote:

The issue of Roma social exclusion only by social work tools is doomed from the start, as social work only renders some symptomatic manifestations invisible, but does not eliminate the deep cause of social exclusion, says Scullion and Brown (2017).

On this same page, you wrote:

From the suggested point of view, there is no choice but to reject how the Czech government as a whole has so far "resolved" the Roma issue, says Albert (2020).

There are many times in your paper when you write that a particular author or authors say something. Please do not do this. In academic writing, you generally write the following:

The issue of Roma social exclusion only by social work tools is doomed from the start, as social work only renders some symptomatic manifestations invisible, but does not eliminate the deep cause of social exclusion (Scullion & Brown, 2017).

On this same page, you wrote:

According to Albert (2020), from the suggested point of view, there is no choice but to reject how the Czech government as a whole has so far "resolved" the Roma issue.

On Page 3, you wrote:

Children should learn about their culture at school.

·        How are you defining culture?

·        What about people that say home (parents) should be children’s primary learning environment?

On Page 3, you wrote:

Kyuchukov at all. (2015) argue, that the second problem is that there are no Roma educators and so the children lack these role models. Roma are increasing in education, there indeed were not that many in secondary schools a decade ago, but as far as education is concerned, they appear in the position of Roma assistants, but how many times do they only have a teaching certificate and are employed to make a point that they have a Roma assistant in a school, informed Bačlija and Grabner (2014).

It should be Kyuchukov et al. (2015) argue the second problem is …..

·        Why are there few Roma educators? Has it always been this way? Provide numbers to support this statement.

·        It is your view that Roma educators are the best OR only appropriate role models for Roma children? If so, provide scholarly support for this view. 

THE AUTHOR MUST CORRECT SEVERAL ERRORS ON THE REFERENCE PAGE. THE PAPER SHOULD BE FORMATTED ACCORDING TO THE 7TH EDITION OF AMERICAN PSYCHOLOGICAL ASSOCIATION (APA)

·        All citations should be in alphabetical order

·        You should capitalize Journal Titles

·        You should italicize Volume Numbers

·        You should italicize book titles

·        You should NOT Italicize Issue Numbers

·        You should provide beginning and ending page numbers

·        You should NOT italicize page numbers

Author Response

Good afternoon,

thank you for your feedback, as we have uploaded a revised version of the article

ree the answers to some of your inquiries below,

reagrds

Mgr. Rudorfer

Children should learn about their culture at school.

  • How are you defining culture?

  • What about people that say home (parents) should be children’s primary learning environment?

 The Council of Europe has called on its member countries to include Roma history and culture in school curricula. According to her, education in this direction will help in the fight against hatred, discrimination and prejudice against this minority. According to the continental organization, the inclusion of Roma history in teaching would increase awareness of the fact that this minority has been "an integral part of national and European societies" for centuries. According to the recommendations, it is important to teach in schools about the Roma Holocaust during the Second World War, which is the responsibility of Nazi Germany and its allies, but also about other acts committed against this minority. According to the Council of Europe, the historical periods in which Roma, Sinti or Irish nomads were not victims should be discussed in schools and outside them. Children should also learn about the positive contribution of the minority to the regional, national and European cultural heritage. The active role of the Roma and Sinti in the anti-fascist resistance should not be forgotten either.

The recommendation encourages member states of the Council of Europe to fight against persistent anti-Roma sentiments, namely through better education in history classes. The teaching of the history of the Roma should also focus on the contribution of this minority in the economic field, especially in the field of trade, blacksmithing or animal husbandry. However, pupils should learn about the Roma and their culture not only in history classes but also, for example, in civics, art and music education. The Roma cultural heritage, including literature, piety, music and various traditions, should not be forgotten either. At the same time, teachers should emphasize the "unequal access to social rights" that this minority has faced for centuries. In order to prevent the emergence of hatred, discrimination and prejudice, teaching in schools should also address the issues of propaganda and false information that are spread on the Internet and social networks in relation to the Roma, the Council of Europe recommended. In recent years, the Council of Europe has repeatedly called on member states, including the Czech Republic, to improve Roma access to education and housing. The Council of Europe was established in 1949 and is based on the intergovernmental cooperation of 47 countries. It deals with the protection of human rights and strives to strengthen democracy and the principles of the rule of law. The Council of Europe is not an institution of the European Union.

Social and cultural specifics are factors that belong to a certain national minority and have an influence on the formation of their identity in society. They are characteristic of her and in a certain way they differ from the majority. They may or may not cause educational complications. However, these specifics do not only concern the Roma minority, but also persons with mental or physical disabilities, the elderly, immigrants or people with addictions. However, if we relate these specifics specifically to the Roma minority, we will have an enormous number of determinants influencing their education. Another determinant can be the education level of the parents or so-called social reproduction. As Němec writes in his publication regarding the issue of the education of Roma pupils: "If we simplify the definition of social reproduction, then we can say that the social status of parents significantly affects (reproduces) the future status of children. So it could be said that the higher the social class the family comes from, the more interested they are in their child's education and future.

On Page 3, you wrote:

Kyuchukov at all. (2015) argue, that the second problem is that there are no Roma educators and so the children lack these role models. Roma are increasing in education, there indeed were not that many in secondary schools a decade ago, but as far as education is concerned, they appear in the position of Roma assistants, but how many times do they only have a teaching certificate and are employed to make a point that they have a Roma assistant in a school, informed Bačlija and Grabner (2014).

It should be Kyuchukov et al. (2015) argue the second problem is …..

  • Why are there few Roma educators? Has it always been this way? Provide numbers to support this statement.

  • It is your view that Roma educators are the best OR only appropriate role models for Roma children? If so, provide scholarly support for this view.  

 A school that wants to successfully educate Roma children must also find a way to reach their parents. they must be the initiator of a process that will help overcome that mutual barrier of mistrust and hostility, and they must convince parents that education is really important for their children. Without cooperation from the family, children's school work cannot have satisfactory results. It is therefore up to the teachers to try to establish a positive relationship with the child's parents from the beginning, and therefore they must, first of all, treat the parents as partners, talk to them not only about school problems but be interested in the whole family, help them in matters where they the parents themselves don't know advice (e.g. more complex dealings with the authorities). Inadequate adaptation of parents to school requirements (unwillingness or inability to provide the child with appropriate school supplies, help him with homework, ensure regular and timely attendance at school) the school should initially be able to forgive and solve with its above-standard care for the children, but talk about the problems with the parents, and gradually increase demands for mutual cooperation. A Roma teacher could help with this.

Teachers should try to get to know and understand the family environment of children, visit families, and learn to know and respect their customs. Parents need to be convinced that the school is not an institution hostile to either the children or the parents. In the vast majority of today's schools, teacher visits to children's families are considered an above-standard but this is not possible for time and economic reasons to happen in real practice. This is, of course, an over-standard, and the issue of measure, possibly also the financial issue, needs to be discussed. Despite all the hard work and demands, however, the effort devoted to convincing parents of the necessity of their children's education is an investment that must pay off in the long run.

link to the fulltext: https://docs.google.com/document/d/1Rtny9MyEJkoZsfQkhMtQJoylKqiQDW7Z/edit?usp=sharing&ouid=109248543351033847896&rtpof=true&sd=true

Round 2

Reviewer 3 Report

After reading the revised manuscript, it is obvious to me that the author made a conscious effort to address all of the issues that I previously raised. Now that the author has made the reader aware of the importance of this topic from the onset (Abstract and Introduction), it is clear why the lack of educational opportunities for Roma children exist, the attitudes and behaviors that encourage these educational realities, as well as cultural interventions that can help remedy this problem. I especially appreciated how the author highlighted the complexities among highly-educated Romani men and women as well as how policy and practitioners can help remedy this problem.